# Crosslinked Polynorbornene-Based Anion Exchange Membranes with Perfluorinated Branch Chains

**DOI:** 10.3390/polym15051073

**Published:** 2023-02-21

**Authors:** Dafu Cao, Xiaowei Sun, Huan Gao, Li Pan, Nanwen Li, Yuesheng Li

**Affiliations:** 1Institute of Advanced Polymer Materials, School of Materials Science and Engineering, Tianjin University, Tianjin 300350, China; 2State Key Laboratory of Coal Conversion, Institute of Coal Chemistry, Chinese Academy of Sciences, Taiyuan 030001, China; 3Collaborative Innovation Center of Chemical Science and Engineering, Tianjin 300072, China

**Keywords:** polynorbornene, crosslinking, anion exchange membrane, perfluorinated, ion conductivity

## Abstract

To investigate the effect of perfluorinated substituent on the properties of anion exchange membranes (AEMs), cross-linked polynorbornene-based AEMs with perfluorinated branch chains were prepared via ring opening metathesis polymerization, subsequent crosslinking reaction, and quaternization. The crosslinking structure enables the resultant AEMs (CFnB) to exhibit a low swelling ratio, high toughness, and high water uptake, simultaneously. In addition, benefiting from the ion gathering and side chain microphase separation caused by their flexible backbone and perfluorinated branch chain, these AEMs had high hydroxide conductivity up to 106.9 mS cm^−1^ at 80 °C even at low ion content (IEC < 1.6 meq g^−1^). This work provides a new approach to achieve improved ion conductivity at low ion content by introducing the perfluorinated branch chains and puts forward a referable way to prepare AEMs with high performance.

## 1. Introduction

Anion exchange membrane fuel cells (AEMFCs) have been very attractive for potentially replacing commercialized but expensive proton exchange membrane fuel cells (PEMFCs) in vehicle, power station, and portable power supply fields because of the high oxygen reduction reaction (ORR) activity and the ability to use low-cost non-noble metal electrocatalysts [1,2,3,4,5,6,7,8]. As the solid electrolyte in AEMFCs, anion exchange membranes (AEMs) are responsible for isolating fuel and oxidants, bearing electrocatalysts and binders, and conducting anions and water [9,10,11]. In the past decade, various polymers with different backbones, such as poly(phenylene oxide) (PPO) [12,13,14], poly(arylene ether sulfone) (PAES) [15,16,17], poly(arylene ether ketone) (PAEK) [18,19,20], poly(benzimidazolium) (PBI) [21,22,23], poly(biphenyl) (PBP) [24,25,26], polyfluorene (PF) [27,28,29], and polyolefin (PO) [30,31,32,33], have been prepared and worked as AEMs to pursue excellent performances in fuel cells application, including high ion conductivity, good alkaline stability, and good dimensional stability.

Among all functional polymers applied for exchange membranes, fluorinated polymers play an irreplaceable role due to their remarkable properties, such as high chemical and thermal stability, hydrophobicity, safety, and processing properties [34,35,36,37,38,39,40]. Some partially fluorinated polymers have already been applied in AEMs. For example, Varcoe et al. [41,42,43,44,45] prepared grafted AEMs with ethylene and tetrafluoroethylene copolymer (ETFE) backbone by radiating the ETFE films with high-energy electron beam and treating the films via quaternarization. The obtained ETFE-based AEMs had high peak power densities (up to 1.6 W cm^−2^) in AEMFCs. However, the radiation grafting method could lead to the degradation of the polymer matrix and sacrifice the mechanical strength of the AEMs [46,47].

Preparing fluorinated polyolefin-based AEMs by compositing PTFE with other polymers is an alternative method [48,49]. Wang et al. [48] in situ copolymerized dimethyl diallyl ammonium chloride and divinylbenzene in the pores of the PTFE matrix to prepare fluorinated AEMs. The obtained AEMs exhibited good dimensional stability in the in-plane direction, and the fuel cells with these composite AEMs had high peak power density reach up to 1.1 W cm^−2^ at 80 °C. However, the pores in the PTFE matrix may reduce the isolation effect of the AEMs on the fuel and oxidizer, and the ionomer part may still be unstable in an alkaline environment.

Synthesizing fluorinated AEMs via direct copolymerization to introduce a few fluorine atoms into the polymer backbone or side chain was a more attractive and widely applicable method [50,51,52,53,54]. For example, Liu et al. [55] synthesized fluorinated poly (arylene ether sulfone)-based AEMs with trifluoromethyl on backbone via polycondensation and subsequent benzylic bromination and quaternization. These obtained AEMs had high hydroxide conductivity (up to 120.5 mS cm^−1^ at 80 °C) and good dimensional stability. However, because of the heteroatoms in the backbone, these AEMs usually suffered from the unstable nature (because of the easily degraded backbone) and could not meet the requirement of stability. In contrast, polyolefins bearing stable carbon–carbon covalent bonds in mainchain presented excellent chemical and electrochemical stability [56,57]. Hickner et al. [54] prepared a series of fluorinated polyolefin-based AEMs via copolymerization of 4-(4-fluorophenyl)-1-butene with 11-bromo-1-undecene and subsequent quaternization. Compared with the contrast sample without fluorine atoms, fluorine-containing AEMs exhibited lower water uptake, higher ion conductivities, better alkaline stability, and much improved fuel cell performance (peak power density of 1.0 W cm^−2^ at 60 °C). These results indicated that the presence of fluorine atoms on polymer backbone or side chain can play a significant role in the performance of AEMs.

In this work, we attempted to investigate the effect of different perfluorinated branch chains on the properties of polyolefin-based AEMs. A series of crosslinked polynorbornene-based AEMs, which could be easily synthesized and functionalized via ring opening metathesis polymerization (ROMP) of polar norbornene derivatives with perfluorinated branch chains by Grubbs third generation catalyst (G3), has been investigated. To clearly reveal the practical effect of perfluorinated branch chains on the newly obtained AEMs, their physical performance, chemical and electrochemical properties were studied in depth.

## 2. Materials and Methods

### 2.1. Materials 

The chemical materials, including 1H,1H,2H-perfluoro-1-hexene (98%), 1H,1H,2H-perfluoro-1-octene (98%), 1H,1H,2H-perfluoro-1-decene (99%) and N,N,N′,N′-tetramethyl-1,6-hexanediamine (TMHDA, 99%) were purchased from J&K Scientific. Allyl bromide (98%), dicyclopentadiene (97%), trimethylamine (TMA), aqueous solution (~4.3 M), and ultradry o-xylene were purchased from Ionochem. Dichloromethane (AR) was purchased from JT Cheme and refined by Etelux SPS800 solvent purification system. The cyclic monomers, 5-(bromomethyl)bicyclo [2.2.1]hept-2-ene (BHE) and 5-(perfluoro-n-alkyl)nobornenes (NBFn, n = 4, 6 or 8) with fluorocarbon chain lengths (n), were synthesized according to the procedure described in the literature [58,59,60]. Grubbs 3rd generation catalyst (G3) was also synthesized according to past report [61].

### 2.2. Synthesis of Random Norbornene Copolymer (FnB)

The typical polymerization process of norbornene random copolymer was as follows (Figure 1): under nitrogen atmosphere, 8 mmol NBFn (n = 4, 6 or 8) and 12 mmol BHE were added into 100 mL glass flask, then 80 mL of dry and degassed dichloromethane was added into the flask under magnetically stirring at −20 °C. Then, 8.8 mg G3 (10 μmol) was dissolved in 10 mL of dry and degassed dichloromethane, then the catalyst solution was quickly poured into the glass flask. After 12 h reaction, 1 mL ethyl vinyl ether was added into the flask to quench reaction. The solution was precipitated in 500 mL methanol and filtered to give white copolymer. The cyclic olefin polymer was dried at 40 °C under vacuum to a constant weight and named as FnB (n = 4, 6 or 8), where n represents the fluorocarbon chain lengths and 60% is the molar content of BHE in copolymer.

### 2.3. Preparation of Crosslinked AEMs with Perfluorinated Branch Chains (CFnB)

As shown in Figure 1, 0.3 g of FnB (n = 4, 6 or 8) was dissolved in 45 mL of o-xylene at 60 °C in a 100 mL glass flask. As the cross-linking reagent, TMHDA (0.05 g/mL in o-xylene) of 1.30 mL (0.38 mmol), 1.11 mL (0.32 mmol) or 0.98 mL (0.28 mmol) for F4B, F6B, or F8B, respectively, was added into the flask. Thereinto, the molar fraction of TMHDA relative to the crosslinkable sites in the copolymer was 50% to theoretically ensure a 100% crosslinking degree. The obtained solution was pre-crosslinked in sealed flask for 3 h under stirring at 60 °C and subsequently poured into a glass Petri dish (9 cm in diameter) at 60 °C for 24 h to give a homogeneous and transparent membrane. The membrane was further immersed into TMA aqueous solution at sealed pressure bottle at 60 °C for 72 h to complete the quaternization of residual bromomethyl groups. After repeatedly rinsing with deionized water and drying in vacuum oven at 60 °C for 12 h, the crosslinked AEM with thickness of 40–50 μm was named as CFnB (n = 4, 6 or 8). AEMs in OH^-^ form were prepared by soaking into 1 M NaOH aqueous solution for 24 h at 25 °C. After immersing, the AEMs were washed with deionized water at least five times to remove residual NaOH thoroughly and dried under vacuum at 60 °C for 12 h.

### 2.4. Characterization and Measurement

^1^H nuclear magnetic resonance (NMR) spectra of copolymers were measured by a Bruker AM-400 instrument (400 MHz for ^1^H) with CDCl_3_ as solvent at rt. Fourier-transform infrared (FTIR) spectra of copolymers and AEMs were recorded by IRTracer-100 spectrometer with a ZnSe crystal in the range of 400–4000 cm^−^^1^. Thermogravimetric analysis (TGA) of AEMs was tested by TA-Q50 analyzer in N_2_ atmosphere from 50 to 600 °C with a ramp rate of 10 °C/min. Before test, samples were preheated at 120 °C for 10 min to remove the residual moisture. Molecule weight and its distribution of copolymers were measured by PL-GPC 50 Integrated GPC system with THF as an eluant at 40 °C at a flow rate of 1.0 mL/min, and PL Easi-Cal PS-1 was used as standard sample for calibration. Tensile properties of dry AEMs were measured by Instron 5969 universal tester at rt with a stretching speed of 20 mm/min. Microstructure of AEMs was conducted via small angle X-ray scattering (SAXS) by Saxesess mc2 small-angle diffraction instrument from Anton Paar with Cu Kα slit collimated X-ray source at 40 kV and 50 mA at 25 °C. 

Ion exchange capacity (IEC) values of AEMs in Br^−^ form were calculated by Mohr’s titration. The weight (*M_dry_*) of a dry AEM was measured and the AEM sample was soaked into 50 mL 1 M NaNO_3_ aqueous solution at rt for 48 h. After that, the solution was titrated with 0.01 M AgNO_3_ aqueous solution using K_2_CrO_4_ as an indicator. The value of IEC can be calculated by following equation: 

This is example 1 of an equation:(1)$$IEC\left({\mathrm{meq}\;g}^{-1}\right)=\frac{V_{AgNO_{3}}\left(\mathrm{mL}\right)\times 0.01\mathrm{mol}/L}{M_{dry}\left(g\right)}$$
where *V_AgNO__3_* represents the consumed volume of AgNO_3_ aqueous solution.

The length (*L_dry_*) and weight (*W_dry_*) of a dry AEM in OH^−^ form were recorded. Then, the AEM was immersed into deionized water at different temperatures (25, 40, 60, or 80 °C) for 12 h, and the length (*L_wet_*) and weight (*W_wet_*) of the saturated AEM were measured quickly. Water uptake (*WU*) and swelling ratio (*SR*) of AEMs were calculated as follows:(2)$$WU=\frac{W_{wet}-W_{dry}}{W_{dry}}\times 100\%$$
(3)$$SR=\frac{L_{wet}-L_{dry}}{L_{dry}}\times 100$$

Ion conductivities of AEMs in OH^−^ form at different temperatures (25, 40, 60, or 80 °C) were measured via alternating current (AC) impedance spectroscopy by Bio-Logic SP-150 impedance/gain phase analyzer with two-point probe technique from 10^6^ to 50 Hz. The ion conductivity was calculated as follows: (4)$$\sigma \left({\mathrm{mS}\;\mathrm{cm}}^{-1}\right)=\frac{l}{R\times d\times h}$$
where *l* is the length of wet AEM sample between the two probes, *R* is the obtained ohmic resistance of AEM sample according to Nyquist plot, *d* and *h* are the width and thickness of wet AEM sample, respectively.

Alkaline stability of AEMs was investigated by immersing AEMs into 1 M NaOH solution at 80 °C for given time. The FTIR spectra of original and aged AEMs in OH^−^ form were contrasted to investigate the change of chemical structures and assess the alkaline stability.

## 3. Results and Discussion

### 3.1. Polymer Synthesis and Characterization

To obtain the AEMs with the desired microstructure, two cyclic monomers, 5-(bromomethyl)bicyclo[2.2.1]hept-2-ene (BHE) and 5-(perfluoro-n-alkyl)nobornenes (NBFn, n = 4, 6 or 8) with varying fluorocarbon chain lengths were first designed and synthesized [58,59,60]. The terminal Br atom in BHE could be further reacted via following quaternization to form a crosslink structure, while the pendant perfluorinated tail of the other cyclic monomer is expected to improve the performances including ion gathering, ion conductivities, and alkaline stability of the AEMs. As shown in Figure 1, polynorbornene-based copolymers with perfluorinated branch chains were firstly synthesized via ROMP of BHE and NBF_n_ with a ratio of 3:2 by using G3. This ratio of two monomers could ensure and reach a good balance that their corresponding AEMs have high ion content and high perfluoroalkyl side chain content, simultaneously, while the increase of perfluoroalkyl side chain content would signally decrease the ion content of AEMs [62,63]. Furthermore, as shown in Table 1, the molecule weight of FnB series copolymers increases with the increase in fluorocarbon chain length of NBFn, and exhibits a low distribution.

Figure 1 presents the ^1^H NMR spectra of these copolymers. The resonances peak (*δ* 5.9–6.3 ppm) of the unreacted double bond of the monomers disappeared after the copolymerization, while double bond (proton named as “a” in the insertion of Figure 1) in the polymer backbone appeared at *δ* 5.0–5.7 ppm, indicating a complete conversion of all monomers. In addition, resonance of the -C*H*_2_Br group (g in the insertion of Figure 1) appears at *δ* 3.1–3.5 ppm. The composition of the copolymers can be easily calculated by the original feed ratio or the ratio of the characteristic integral area of two monomers according to the ^1^H NMR spectra of the copolymers, which both come to very similar results.

After subsequently crosslinking with N,N,N′,N′-tetramethyl-1,6-hexanediamine (TMHDA) and quaternarization with trimethyl amine (TMA), corresponding crosslinked AEMs (CFnB, n = 4, 6, or 8) were obtained. Because the crosslinked chemical structures of CFnBs could not be clearly determined by liquid NMR due to their poor solubility, FTIR spectra of these AEMs were collected and shown in Figure 2. 

As observed, strong and sharp absorption peaks corresponding to the stretching vibration and deformation vibration of C–F bonds appear around 1200 and 1145 cm^−1^, respectively. The *v*(C-N^+^) stretch vibrational absorption peak of CFnB presents at about 908 cm^−1^, indicating that the copolymers have undergone a quaternization reaction.

### 3.2. Ion Exchange Capacity, Water Uptake, and Swelling Ratio

As shown in Table 1, theoretical IEC values for these crosslinked AEMs were calculated by assuming a 100% crosslinking degree between the copolymer and the crosslinking agent TMHDA, and the practical IEC values for the crosslinked AEMs were tested by Mohr’s titration. The practical IEC values of CFnB series AEMs decrease with the increase in fluorocarbon chain lengths. CF4B with the shortest fluorocarbon chain has the highest practical IEC value as 1.52 meq/g, while CF8B with the longest fluorocarbon chain has the lowest practical IEC value as 1.20 meq/g. The practical IEC values of the AEMs are between 73 and 78% of the corresponding theoretical values, indicating that crosslinking and quaternization under heterogeneous conditions are incomplete, which is believed to be the major reason for the low IEC values of CFnB series AEMs [53].

For the crosslinked AEMs, the crosslinked structure could help to inhibit the swelling ratio of the saturated AEMs and endow them with good dimensional stability [33,63]. Therefore, the crosslinked AEMs with higher ion content could still maintain its morphology at high temperature and show higher ion conductivity. The relationship between the water uptake or swelling ratio with the temperature of each AEM is shown in Figure 3. For all AEMs, the WU and SR increase with the increase in the temperature, showing an evidently positive correlation trend. Therefore, CF8B with the lowest IEC value and the longest fluorocarbon chain shows the lowest WU (185.3% at 80 °C) and SR (50.4% at 80 °C), while CF4B with the highest IEC value and the shortest fluorocarbon chain has the highest WU (230.0% at 80 °C) and SR (59.0% at 80 °C). All AEMs could still maintain low SRs even with high WUs, which indicates that the crosslinking structure and the perfluoroalkyl branch chain both contribute to a good dimensional stability of the obtained AEMs.

### 3.3. Thermal and Tensile Properties

In AEMFCs, AEMs usually operate at elevated temperatures (60–120 °C) and therefore require good thermal stability [3]. Figure 4 compares the thermogravimetric curves of the obtained AEMs and all the AEMs have two main degradation steps. As observed, after the crosslinking and quaternization reactions, CF4B exhibits a much lower thermal decomposition temperature (T_d,95_s) than F4B, indicating that the first degradation stage of AEMs, from 170 to 290 °C, corresponds to the deamination of side chains with quaternary ammonium cation tethered on the backbone [64,65]. The second degradation stage in the range of 290–480 °C is related to the degradation of the main chain and the perfluorinated branch chain. Furthermore, the weight loss percentages of AEMs are given in Table 2. The weight loss percentage of the first degradation stage decreased with the decrease in ion content of the AEMs, supporting the conclusion above. In general, the thermal decomposition temperatures of these AEMs are all above 210 °C, which can meet the requirements for practical operation of AEMs in AEMFCs.

High mechanical strength is essential for AEMs to maintain the dimension stabilizing in fuel cells [25,30]. However, AEMs based on polyolefins usually suffer from low mechanical strength, and additional strategies such as crosslinking or introducing rigid large steric hindrance substituent groups are necessary [30,65,66]. The tensile test results and representative tensile curves of crosslinked CFnB series AEMs are conducted and shown in Table 3 and Figure 5, respectively. The tensile strength of the AEMs is in the range of 14.1–15.5 MPa, Young’s modulus is in the range of 375–497 MPa, and elongation at break is in the range of 13.8–98.8%. Thereinto, CF6B with a moderate fluorocarbon chain length has the highest Young’s modulus (15.5 MPa) and tensile strength (497 MPa). Furthermore, it is worth noting that all AEMs have high tensile strength (>14 MPa) and good toughness, which will both be beneficial for their dimensional stability in fuel cells.

### 3.4. Microstructure Characterization

It has been widely accepted that evident microphase separation or well-defined ionic clusters will increase the water uptake and promote the ion transportation of AEMs [67,68,69,70]. Here, the microphase-separated morphology of CFnB series AEMs were studied by small angle X-ray scattering (SAXS). As shown in Figure 6, all AEMs have a clear scattering peak near 4.1 nm^−1^, and the corresponding characteristic separation length (*d*) is about 1.5 nm according to the equation *d* = 2π/*q*. This scattering peak should be caused by the aggregation of the quaternary ammonium cations. In addition, with the increase in the fluorocarbon chain length, a scattering peak becomes more distinct at about 1.4 nm^−1^, and the corresponding characteristic distance *d* is about 4.5 nm. This phenomenon indicates that this peak represents the microphase separation of perfluorinated branch chains. In general, even with the crosslinking structure, all AEMs have a distinct ion gathering because of the flexible backbone, and the perfluorinated branch chain contributes to the formation of microphase separation [71,72,73,74].

### 3.5. Ion Conductivity

The hydroxide conductivities of CFnB series AEMs from 25 to 80 °C were shown in Figure 7. Therefore, CF4B with the highest IEC value (1.52 meq g^−1^) has the highest hydroxide conductivity, which is 58.3 mS cm^−1^ at 25 °C and 106.9 mS cm^−1^ at 80 °C. The hydroxide conductivity of CF6B with a little lower IEC value (1.41 meq g^−1^) is slightly lower than that of CF4B, which is 56.2 mS cm^−1^ at 25 °C and 98.1 mS cm^−1^ at 80 °C. Even though the IEC value is only 1.20 meq g^−1^, CF8B with a perfluorinated branch chain also has a hydroxide conductivity of 50.1 mS cm^−1^ at 25 °C and 88.4 mS cm^−1^ at 80 °C, indicating that ion gathering and the side chain microphase separation caused by a flexible backbone and perfluorinated branch chain contribute to ion transport [75,76,77]. Furthermore, higher WU is also supposed to benefit the high ion conductivities of CFnB series AEMs at low IEC values. Figure 8 compares the IEC values and hydroxide conductivities of crosslinked AEMs in this work and the previous literature [14,63,65,78,79,80]. As observed, even with a low ion content, CFnB series AEMs with perfluorinated branch chains tend to exhibit higher hydroxide conductivities than those of crosslinked AEMs with much higher ion contents. This result further indicates that the introduction of perfluorinated branch chains could greatly promote ion transport.

### 3.6. Alkaline Stability

When operating in fuel cells, the alkaline stability of AEMs in a hot alkaline environment signally affects the durability and lifetime of fuel cells. Here, we soaked the CFnB series AEMs in 1 M NaOH aqueous solution for 16 days at 80 °C, and FTIR spectra of the AEMs were compared to reveal the variation in chemical structure of the AEMs. As shown in Figure 9, absorption peak in the range of 1600–1700 cm^−1^ of all AEMs decreased significantly, which corresponds to the degradation of the C=C bond on the polynorbornene backbone. In addition, the absorption peaks of AEMs in other regions did not show evident change. These results indicate that the perfluorinated branch chains would not damage the alkaline stability of CFnB series AEMs.

## 4. Conclusions

In this work, crosslinked polynorbornene-based AEMs (CFnB) with perfluorinated branch chains were prepared via ROMP of 5-(bromomethyl)bicyclo[2.2.1]hept-2-ene and 5-(perfluoro-n-alkyl)nobornenes (n = 4, 6, 8) and subsequent crosslinking as well as quaternarization. Under the combined effect of the crosslinking structure and the perfluorinated branch chains, the CFnB series AEMs exhibited a inhibited swelling ratio, high tensile strength, and a high water uptake, indicating their good dimensional stability. Even with low ion content (IEC < 1.6 meq g^−^^1^), all CFnB series AEMs had high hydroxide conductivities. CF4B with IEC value as 1.52 meq g^−^^1^ had the highest hydroxide conductivity at 80 °C. The ion gathering and side chain microphase separation caused by a flexible backbone and perfluorinated branch chain, as well as a high water uptake, were believed to promote the ion conductivities together. This work provides a new attempt to achieve higher ion conductivity at a low ion content, and puts forward a referable way to prepare AEMs with high performance for fuel cells application. In our next works, increasing the ion content of CFnB series AEMs with perfluorinated branch chains and hydrogenating the unsaturated polynorbornene backbone, thus preparing AEMs with higher ion conductivity and alkaline stability, would be the direction of efforts. 

## Data Availability

The data are available on reasonable request from the corresponding author.

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
