# Peer review of "Crosslinked Polynorbornene-Based Anion Exchange Membranes with Perfluorinated Branch Chains"

_polymers, 2023, doi:10.3390/polym15051073_

Round 1

Reviewer 1 Report

Ion-exchange membranes with perfluorinated side chains have been prepared by a novel method. Due to microphase separation, the membranes have good conductivity. Authors’ experimental data showed good results. But there are still some points that authors need to pay attention to improve the quality of the paper. Please see the comments below.

Page 2 line 49, PTFE matrix have been successfully used in the fabrication of reinforced PEM and have shown good fuel cell performance. This laid the foundation for the application of PTFE matrix in the manufacture of reinforced AEM. Please refer "Poly (phenylene sulfonic acid)-expanded polytetrafluoroethylene composite membrane for low relative humidity operation in hydrogen fuel cells." Journal of Power Sources 535 (2022): 231375.

Page 3 line 116, In my experience, 0.1M NaOH soaking may give AEM better performance. High concentration of hydroxide may cause partial degradation of the membrane

Page 4 line 146, Measurement of conductivity with two electrodes is not as accurate as that with four electrodes

Page 4 line 149, Is the cross-sectional area from dry membrane or wet membrane?

Page 8 Figure 5, Could authors explain why CF6B has the highest tensile strength?

Page 9 line 271 It’s better to replace ‘ion content’ by ‘IEC’

Reviewer 2 Report

In this manuscript, the authors reported three new AEMs based on fluorinated polynorbornene. Detailed characterization and analysis, including synthesis, IEC, WU, SR, thermal properties, tensile properties, microstructure, and alkaline stability, were provided.  The fluorinated branched chain helped to improve the dimensional stability and conductivity for AEMs. My recommendation is that it is a very important paper to be published in Polymers after minor revision. Here are some suggestions:

1.       In section 3.1, the molecular weight of each FnBr should be provided because molecular weight has remarkable impact on other parameters, like WU, SR, mechanical properties, conductivity and cell performance.

2.       In Fig 2, the IR spectrum of CF4B is different from the spectrums of CF6B and CF8B. The strong peaks around 650 and 570 cm-1 are missing, while the peak around 1200 cm-1  is almost covered by the peak around 1250 cm-1.  If the peak around 650 cm-1 is assigned to the C-Br bond, there will be a lot of bromide groups in CF6B and CF8B. Then it is not so reasonable for the gap of practical IEC and theoretical IEC for these three AEMs. In section 2.3, TMA aqueous solution was heated at 60 °C for 72 h, while the TMA is very easy evaporated. Did the authors replenished the TMA solution? The peak at 1646 cm-1 should be assigned to C=C bond instead of water.

3.       In section 3.3, reference for the analysis of the degradation reasons should be provided if the authors are sure that the first stage is the degradation of cation group, the second stage is of main chain and fluorinated branch chain. The C-F bond should be easily abstracted by the nearby C-H bond (marked as h in Fig 1). Also, the cation group is easily degraded, too. The weight loss percentages should be also provided here. Two good references could be cited here: Journal of Membrane Science 2018, 556, 118.; Int. J. Hydrog. Energy 2016, 41. 16264.

4.       In section 3.6, the authors claimed that the only obvious change in FTIR was around 1600-1700 cm-1, which resulted from the degradation of C=C bond on the polynorbornene in 1M NaOH at 80 °C after 16 days. Is it possible that the reason is the formation of new C=C bond due to the abstraction of C-F bond by the nearby C-H bond (marked as h in Fig 1)? To rule out this possibility, the authors could test the alkaline stability of FnB under the same conditions (reference: The Journal of Organic Chemistry 2021, 86, 254)

5.       The conclusion that “These results indicated that the presence of fluorine atoms on polymer backbone can play a significant role on the performance of AEMs” is not very accurate. Because the cited references show the important roles of the fluorine atoms on backbone or branch chain, not only on backbone. And here, it will be a more professional point of view if the authors could mention the current issue in AEM based on polyolefin, like, low mechanical properties.

6.       In section 2.3, on page 3 line 106, “CFnB” should be “FnB”. On page 3 line 108, the appropriate amount of TMHDA should be provided directly, instead of the explanation of ratio between FnB and TMHDA. As there is high requirement for the ratio of TMHDA to FnB and the volatility of organic liquid compounds is high, the authors should emphasize that the reaction flask was sealed during the pre-crosslinked step for 3 h under stirring at 60 ºC.

7.       There is no symbol for x(QH)QPPO-60 [77] in Fig 8.

8.       The reported AEM in ref 30 is not polyfluorene (PF), but poly(arylene ether sulfone).

9.       The reported polymer in ref 36 is not fluorinated polymers.

10.    The reported composited membrane in ref 49 is not AEM, but PEM in DMFCs.

11.    The reported AEM in ref 51 is only a copolymerization of polyolefin, not a fluorinated polymer. Two fluorinated polyolefins should be cited here: J. Membr. Sci. 2019, 574, 212.; Adv. Funct. Mater. 2019, 29, 1902059.

12.    The reported AEM in ref 52 is PEM, not AEM.

13.    There are some grammatical mistakes. On page 5 line 184, the sentence that “In consideration of the chemical structures of crosslinked CFnBs couldn’t be clearly determined by liquid NMR because of their poor solubility” should be revised as “Because the crosslinked chemical structures of CFnBs couldn’t be clearly determined by liquid NMR due to their poor solubility”. On page 6 line 217, “indicates that” should be revised as “which indicates that”. On page 8 line 249, “which corresponding characteristic separation length (d) according to the equation d = 2π/q is about 1.5 nm.” should be revised as “and the corresponding characteristic separation length (d) is about 1.5 nm according to the equation d = 2π/q”.

Round 2

Reviewer 1 Report

The author has modified the article as suggested, and my suggestion is to accept it.